# Mycomedicine: A Unique Class of Natural Products with Potent Anti-tumour Bioactivities

**DOI:** 10.3390/molecules26041113

**Published:** 2021-02-19

**Authors:** Rongchen Dai, Mengfan Liu, Wan Najbah Nik Nabil, Zhichao Xi, Hongxi Xu

**Affiliations:** 1School of Pharmacy, Shanghai University of Traditional Chinese Medicine, Shanghai 201203, China; ycy220823@gmail.com (R.D.); mengfanliush@163.com (M.L.); najbah@yahoo.com (W.N.N.N.); 2Pharmaceutical Services Program, Ministry of Health, Selangor 46200, Malaysia; 3Shuguang Hospital, Shanghai University of Traditional Chinese Medicine, Shanghai 201203, China

**Keywords:** mycomedicine, medicinal fungi, anti-cancer, polysaccharides, triterpenoids

## Abstract

Mycomedicine is a unique class of natural medicine that has been widely used in Asian countries for thousands of years. Modern mycomedicine consists of fruiting bodies, spores, or other tissues of medicinal fungi, as well as bioactive components extracted from them, including polysaccharides and, triterpenoids, etc. Since the discovery of the famous fungal extract, penicillin, by Alexander Fleming in the late 19th century, researchers have realised the significant antibiotic and other medicinal values of fungal extracts. As medicinal fungi and fungal metabolites can induce apoptosis or autophagy, enhance the immune response, and reduce metastatic potential, several types of mushrooms, such as *Ganoderma lucidum* and *Grifola frondosa*, have been extensively investigated, and anti-cancer drugs have been developed from their extracts. Although some studies have highlighted the anti-cancer properties of a single, specific mushroom, only limited reviews have summarised diverse medicinal fungi as mycomedicine. In this review, we not only list the structures and functions of pharmaceutically active components isolated from mycomedicine, but also summarise the mechanisms underlying the potent bioactivities of several representative mushrooms in the Kingdom Fungi against various types of tumour.

## 1. Introduction

Currently, cancers contribute to the second leading cause of death worldwide and involve a series of complex pathogenic mechanisms [1]. Despite the considerable advancements in anti-cancer therapy, serious problems persist. Chemotherapy and radiation therapy, as examples, remain the main strategies for cancer management. Nevertheless, the effectiveness of synthetic anti-cancer agents for chemotherapy is often limited owing to their serious toxic effects, affecting the metabolism and proliferation of normal tissues, and reducing the quality of life of patients. Therefore, alternative therapeutic approaches for cancer treatment are urgently required.

Since ancient times, the medicinal values of natural products have been acknowledged. A wide range of natural products, including plant- and marine-organism-derived agents, are gaining momentum in anti-cancer therapy owing to their unique advantages of high efficiency and minimal side effects [2]. Mycomedicine, a unique class of natural medicines that includes numerous edible mushrooms and medicinal fungi, as well as their metabolites, has been widely employed for treating different kinds of diseases for thousands of years according to Shen Nong’s Materia Medica [3]. In China, six medicinal fungi have been included in the China Pharmacopoeia, including *Ganoderma lucidum*, which has been extensively researched and has many clinical applications, especially as an anti-cancer agent (Figure 1). Modern technologies have identified that pharmaceutically active compounds isolated from mycomedicine predominantly include triterpenoids, polysaccharides, and proteins. The active ingredients and anti-cancer properties of some of the medicinal fungi have been well elaborated in recent review articles [4,5,6,7,8]. However, a detailed review of numerous anti-tumour bioactivities of various mycomedicines is lacking. In the current review, we discuss nearly 20 kinds of typical mycomedicines worldwide that have exhibited promising bioactivities against tumors (Table 1). More importantly, we provide a detailed summary of the mycomedicine-derived bioactive components, as well as their underlying anti-cancer mechanisms, such as those involved in regulating cancer cell cycle arrest, apoptosis, autophagy, the immune response, metastasis, angiogenesis, oxidation, the gut microenvironment, and multidrug resistance of multiple famous mycomedicines (Table 2, Figure 2). Furthermore, we discuss the current clinical evidence of mycomedicine in anti-cancer therapy.

## 2. Bioactive Components Isolated from Mycomedicine

### 2.1. Polysaccharides

Polysaccharides are long-chain sugar molecules containing monosaccharide units connected via glycosidic linkages. The chemical structures of polysaccharides derived from mycomedicine vary depending on the isolation methods, and marginally differ when extracted from fruit bodies, cultured mycelia, or culture broth [19]. Considering *Grifola frondosa* as an example, all polysaccharides detected in fruiting bodies were *β*-glucans; however, heteromannans, heterofucans, and heteroxylans, and their complexes with protein were detected in its culture mycelium [20]. The core chemical structure of mushroom polysaccharides is presented as *β*-glucans with different glycosidic linkages, including *β*-(1→3) linkages and *β*-(1→6) branch points [21]. Furthermore, mushroom polysaccharides are presented as polysaccharide-conjugate complexes called heteroglucan or polysaccharopeptide (PSP), presenting *α*(1–4)- and *β*(1–3)-glycosidic linkages with protein components. For example, extracts isolated from *Agaricus blazei* contain diverse polysaccharide-protein complexes with different chemical linkages, including *β*-1,6-glucan, *α*-1,6- and *α*-1,4-glucan, glucomannan, and *β*-1,3-glucan [22].

Numerous studies have revealed that polysaccharides are the most widely extracted active ingredients in mycomedicine, demonstrating anti-cancer properties (Table 2) [21]. The anti-cancer properties of polysaccharides reportedly depend on various factors, including sugar composition, molecular weight, water solubility, and glucose linkage [23]. It has been confirmed that the number and length of short branched chains in mushroom polysaccharides can significantly influence their anti-tumour activities [24]. For example, *β*-d-glucans composed of (1→3)-, (1→4)-, and (1→6)-*β*-d linkages, isolated from *G. lucidum*, demonstrated more potent anti-cancer activities and better absorption than other polysaccharides [25]. Furthermore, polysaccharides with high molecular weights exhibit more robust anti-tumour efficacy than those with low molecular weight. For instance, *Coriolus versciclor* PSP and polysaccharide Kureha (PSK) [26], as well as various kinds of polysaccharides extracted from *G. lucidum*, exhibited promising anti-cancer effects with a molecular weight of approximately 10^5^ to 10^6^ Da [27]. Similarly, the highest activity of *Grifola frondosa* polysaccharides was ascribed to one demonstrating a molecular weight exceeding 800 kDa [28].

### 2.2. Terpenes and Terpenoids

Terpenes and terpenoids are composed of linked isoprene units and are generally classified as monoterpenes, diterpenes, and triterpenes, depending on the number of carbon atoms generated via biosynthetic pathways [29]. Higher basidiomycetes or medicinal mushrooms are considered the most important sources of bioactive terpenoids in nature, demonstrating a wide range of therapeutic and curative properties.

Triterpenes and triterpenoids, the largest subclass of natural terpenoids, represent some of the most potent bioactive compounds that contribute to the anti-cancer activities of various medicinal mushrooms (Table 2). Structurally, triterpenes from natural sources have a wide variety of different skeletons, broadly divided according to the number of presented rings. Reportedly, over a hundred kinds of triterpenoids have been identified in *G. lucidum* [30]. Among them, dozens of ganoderic acids, including ganoderic acid A, D, DM, and T, have been isolated, characterised and structurally categorised under lanostane-type triterpenes [31]. In addition to ganoderic acids, a series of ganoderma alcohols were detected in a wide range of triterpenoids isolated from *G. lucidum*. The anti-cancer mechanisms of triterpenoids and triterpenoid-enriched extracts from *G. lucidum* have been well summarised elsewhere [32]. Similarly, lanostane-type triterpene acids were also suggested to be major medicinal components of *Poria cocos*, expected to be potential cancer chemopreventive agents.

Sesquiterpenes, are another major class of terpenes extracted from mycomedicinal agents, generally demonstrating 15-carbon backbone structures constructed with three isoprene units. A range of bioactive sesquiterpenoids, isolated from Flammulina velutipes, are cytotoxic to HeLa cervical, HepG2 liver and KB mouth cancer cells [33]. Irofulven is a semisynthetic clinical anti-cancer drug, derived from the sesquiterpene, illudin-S, extracted from *Omphalotus illudens*. Irofulven has been approved for the treatment of renal cell carcinoma and ovarian cancer owing to its capacity to interfere with DNA replication in cancer cells [34].

### 2.3. Proteins and Amino Acids

Fungi contain several different proteinaceous materials, comprising over 30% of the mushroom dry weight. Protein extracts can be isolated from almost all types of fungi and fungal materials [35]. The anti-cancer effects of mushroom-derived proteins or peptides are closely linked to their amino acid contents and sequences. However, the extracted quantities of mycomedicine-derived amino acids largely differ when taking into account the species and preservation methods [36]. A variety of amino acids have been extracted from medicinal mushrooms, including both essential and nonessential amino acids. For example, agaritine, a soluble fraction of a hot-water extract of *A. blazei*, is a glutamic acid conjugated with a phenylhydrazine. This non-proteinogenic L-alpha-amino acid demonstrated potent anti-tumour activity in leukemic cells [37]. Moreover, special attention should be paid to arginine, a conditionally essential amino acid that contains an *α*-amino group, an *α*-carboxylic acid group, and a side chain consisting of a 3-carbon aliphatic straight-chain ending in a guanidino group [38]. This amino acid demonstrated superior inhibition efficacy against tumour cell growth and could minimise the risk of cancer metastasis [39].

Fungal immunomodulatory proteins (FIPs) are a group of proteins found in a wide variety of mushrooms, possessing highly similar amino acid sequences and significant therapeutic potential. Structurally, FIPs are dimers and share a dumbbell-shaped structure, which is similar to the variable region of immunoglobulin heavy chains [40]. Typically, FIPs consistently exist in low quantities in their original mushrooms, resulting in major limitations regarding their research and application. However, modern techniques have enhanced the production of recombinant FIPs in other organisms such as yeast and bacteria. Intriguingly, recombinant FIPs showed superior activities when compared with those directly extracted from medicinal mushrooms [41].

Lectins are another unique group of proteins or glycoproteins. These are mostly extracted from mycomedicines and play an important role in the regulation of the immune response. They possess at least one non-catalytic domain, which binds reversibly to specific monosaccharides or oligosaccharides. This group of proteins can recognise and interact with various cell-surface carbohydrates/glycoproteins and possess strong anti-proliferative properties against cancer cells [42]. Moreover, lectins from basidiomycetes have diverse features in terms of their physicochemical characteristics and carbohydrate specificity. The possible role of lectins isolated from medicinal mushrooms as cancer therapeutics and the underlying anti-cancer mechanisms of lectins in vitro and in vivo have been well summarised previously [43,44].

### 2.4. Other Bioactive Compounds

Besides the above predominant bioactive constituents, extracts that are derived from mycomedicine also present other types of structures, including nucleosides and phenols. Although the mycomedicine-derived nucleosides garner less attention, these compounds play important roles in regulating various physiological processes in human. Structurally, nucleosides consist of a 5-carbon sugar (either ribose or deoxyribose) and a nucleobase (also termed as a nitrogenous base). Inosine, guanosine and adenosine are the major nucleosides, which account for 60% of the total nucleosides in *Ganoderma* species [45]. Among the numerous nucleosides isolated from the *Cordyceps* family, cordycepin, with structural formula of 3′-deoxyadenosine, demonstrated marked anti-tumour bioactivities [46]. The chemical structures of cordycepin and nucleoside adenosine are very similar, with cordycepin lacking a 3′-hydroxyl group at the ribose moiety that is necessary for 5′–3′ elongation of nucleotide polymers [47]. Clitocine (6-amino-5-nitro-4-(*β*-d-ribofuranosylamino)pyrimidine), an exocyclic amino nucleoside that exerts potent anti-tumour bioactivities in several in vitro and in vivo cancer models [48]. Clitocine was first isolated from *Clitocybe inversa* in the 1980s and is also extractable from other mushrooms, such as *Leucopaxillus giganteus* [49].

In addition to the nucleosides, phenols, as well as fatty acids, vitamin, steroids, and essential minerals are examples of other types of compounds that are present at low quantities in mycomedicines [50]. Several studies have documented the total contents of phenolic compounds derived from selected medicinal mushrooms [51,52,53]. For example, suillin derived from *Suillus* species, is a tetraprenylphenol with a diterpenic chain linked to the aromatic ring in the 3-position [54]. Although these compounds are important groups of secondary metabolites from fungal fruiting bodies, limited studies have explored their anti-tumour bioactivities.

## 3. Major Anti-tumour Bioactivities of Mycomedicine

### 3.1. Induction of Cell Cycle Arrest

In normal mammalian cells, the cell cycle is divided into four phases (G0/G1, S, G2, and M), which are highly regulated by specific complexes of cell cycle regulators, including cyclins and cyclin-dependent kinases (CDKs). However, alterations in cell cycle regulators, especially overexpression of different types of cyclins and CDKs, could consistently facilitate cell proliferation and contribute to neoplastic development. Therefore, targeting these abnormal regulators may provide a possible cancer therapeutic intervention. Studies have demonstrated that an *A. blazei* extract induced G2/M arrest, partially by downregulating the modulation of the cyclin B/cdc2 complex [55] Similarly, WEES-G6, triterpene-enriched extracts isolated from *G. lucidum*, delayed cell division at the G2/M phase via deficiency of cyclin B in Huh-7 hepatoma cells [56]. Furthermore, Liao and colleagues revealed that cordycepin isolated from *Cordyceps militaris* facilitated S-phase arrest by inducing a reduction in the protein levels of cyclin A2, cyclin E, and CDK2 [14].

Alternatively, unfavourable cell cycle progression is delayed and halted by CDK inhibitors (CKIs). The elevated levels of CKIs tightly restrain CDK activity, which could be a promising approach to regulate cancer cell proliferation. The CIP/KIP family, which consists of p21^Cip1^, p27^Kip1^, and p57^Kip2^, is one class of CKI involving the binding and inhibition of G1/S and S-phase cyclin-CDK complexes. Studies have shown that *G. lucidum* triterpene extracts induced cell cycle arrest at the G1 phase by increasing the protein level of p21 and decreasing cyclin D1, CDK4, and E2F in MCF-7 breast cancer cells [57,58]. Additionally, *Ganoderma tsugae* extracts downregulated the protein levels of cyclin A and B1 and upregulated those of p21 and p27, leading to cell accumulation in the G2/M phase in Colo205 colon cancer cells [59].

Moreover, cell cycle arrest could also be induced by regulating CDK substrates. Retinoblastoma protein (Rb) is phosphorylated by activation of CDK4/6-cyclin-D complexes, resulting in the transcription of E2F, subsequently enabling cell cycle progression from the G1 to S phase [60]. Wu et al. demonstrated the downregulation of phosphorylated Rb as well as CDK2, CDK6, cyclin D1, and c-Myc oncoprotein, after treatment with ganoderic acid DM (GADM) [10]. Consistently, Hsu et al. reported that a combination of extracts from *Trametes versicolor* and *G. lucidum* markedly decreased protein levels of phospho-Rb (Ser807/Ser780/Thr821), as well as E2F, impeding the transition from the G1 to S phase in HL-60 leukemia cells [61].

### 3.2. Induction of Apoptosis

Apoptosis, a type of programmed cell death, serves as a barrier to cancer development. Induction of apoptosis offers promising therapeutic potential for cancer treatment. It has been widely documented that there are two major pathways of apoptosis: the extrinsic pathway mediated by cell death receptors, and the intrinsic pathway mediated by mitochondria. Diverse medicinal fungi exert anti-cancer effects by inducing apoptosis through distinct pathways.

The extrinsic pathway of apoptosis is triggered by the binding of the death ligand to its death receptor, subsequently inducing cancer cell death by facilitating the activation of procaspase-8 to cleaved-caspase-8. A significant reduction in c-FLIP protein levels, with subsequently augmented cleaved-caspase-8, was observed in H460 lung cancer cells following treatment with peptides extracted from *Lentinus squarrosulus* [62]. Suillin, extracted from the mushroom *Suillus placidus*, upregulated both death receptor Fas and its adaptor, Fas-associated death domain protein (FADD), with the resultant caspase 8 inducing apoptosis in HepG2 liver cancer cells [15]. Furthermore, ethanol extracts of *G. lucidum* (EGL) displayed apoptotic potency in human gastric carcinoma cells. The protein levels of death receptor 5 (DR5) and tumour necrosis factor-related apoptosis-inducing ligand (TRAIL) were increased by EGL in a concentration- and time-dependent manner, further inducing caspase-8 mediated extrinsic apoptosis [63].

The intrinsic pathway of apoptosis is initiated through mitochondria, closely regulated by the Bcl-2 family and the release of pro-apoptotic molecules into the cytoplasm, subsequently promoting caspase activation [64]. Agaritine, a hydrazine-containing compound extracted from *A. blazei*, triggered apoptosis by activating caspase-8 and -9, as well as the release of cytochrome c from the mitochondria into the cytoplasm, in U937 leukaemic cells [9]. Zhang et al. reported that JLNT, a purified polysaccharide isolated from *Lentinus edodes*, showed potential anti-cancer effects by activating intrinsic apoptotic pathways in vivo, as evidenced by the upregulated expression of tumour suppressor p53, increased the ratio of Bax/Bcl-2 and induction of cleaved-caspase-3, in BALB/c mice bearing S180 xenografts [65]. Shang’s group revealed that a polysaccharide extracted from Se-enriched *G. lucidum* exhibited cytotoxic effects, with typical apoptotic characteristics, including increased activities of caspase-9 and -3 and poly (ADP-ribose) polymerase, in MCF-7 breast cancer cells. Moreover, the inhibitory function of the extract could be significantly reversed by a caspase-9 inhibitor (z-LEHD-FMK) and caspase-3 inhibitor (z-DEVD-FMK) [66]. In addition to caspase-dependent apoptosis, intrinsic apoptosis can be triggered through a caspase-independent pathway. Poricotriol A, isolated from *P. cocos*, triggered apoptosis in A549 lung cancer cells via translocation of apoptosis-inducing factor (AIF) and possibly EndoG, suggesting its anti-tumour effect was exhibited through a caspase-independent pathway [16]. Furthermore, mycomedicine has demonstrated its apoptosis-induction abilities through the endoplasmic reticulum stress (ERS) pathway. Marmorin, isolated from the fresh fruiting bodies of *Hypsizigus marmoreus*, triggered the ERS apoptotic pathway by upregulating phospho-IRE1α, cleaved-caspase-12, and C/EBP homologous protein (CHOP) in both MCF7 and MDA-MB-231 breast cancer cells. Consistent with in vitro results, marmorin significantly induced ERS and DNA damage in an MDA-MB-231 xenograft mouse model presenting elevated protein expression levels of ERS markers and TUNEL-positive cells, when compared with the vehicle control [67]. Collectively, these findings provide a clear insight into the potential value of mycomedicine as a pro-apoptosis candidate for anti-cancer drugs (Figure 2).

### 3.3. Induction of Autophagy

Autophagy, an evolutionarily conserved mechanism, is a critical cellular pathway that maintains cellular homeostasis by degrading unnecessary proteins and organelles in lysosomes for recycling. In cancer, the roles of autophagy have been referred to as a double-edged sword. Autophagy not only activates and inhibits cell death, but also facilitates cell death mechanisms [68]. Emerging evidence has suggested that appropriate pharmacological modulation of autophagy is a promising strategy in cancer therapy.

During autophagy induction, LC3-I is conjugated with phosphatidylethanolamine to form the lipidated LC3-II, which is critical for autophagosome degradation, substrate recognition, and efficient autophagosome closure and fusion with lysosomes [69]. Thus, the ratio of LC3-II/LC3-I has been widely used to monitor autophagic activity [70]. In gastric carcinoma AGS cells, the protein level of LC3-II was significantly augmented after treatment with a methanolic extract of the *G. lucidum* fruiting body [71]. Similarly, increased LC3-II protein levels were observed in the Hep3B xenograft nude mice model of human hepatocellular carcinoma, after treatment with cold-water extracts of *G. frondosa* [72]. The triterpenes isolated from *G. lucidum* (GLT) exhibited a significant inhibitory effect on colon cancer both in vitro and in vivo by inducing autophagy-mediated programmed cell death. GLT induced the formation of autophagic vacuoles and upregulated the protein expression of Beclin-1 and LC-3, at least partially through the suppression of p38 mitogen-activated protein kinase (MAPK) phosphorylation in colon cancer cells and tumour tissues of a xenograft model [73]. The recombinant fungal immunomodulatory protein GMI was cloned and purified from *Ganoderma microsporum* and induced lung cancer cell death and xenograft tumour growth inhibition by robustly activating autophagy but did not induce apoptotic cell death [74]. Calcium-mediated signalling pathways and the autophagy-related genes, LC3, BECN1, and ATG5, have been confirmed to be essential for GMI-induced autophagy.

The final steps of autophagy are the fusion of autophagosomes with lysosomes and degradation of cellular macromolecules in lysosomes. Inhibition of these processes impairs autophagic deregulation, ultimately causing cell survival failure. Qi and colleagues reported that polysaccharides extracted from *Cordyceps sinensis* suppressed the formation of autophagolysosomes, resulting in a blockage of autophagy flux via the mammalian target of rapamycin (mTOR) signalling pathway in colorectal cancer (CRC) cells [75]. Furthermore, the *G. lucidum* polysaccharide (GLP) significantly blocked autophagic flux by inhibiting autophagosome and lysosome fusion in CRC cells. Additionally, a study has revealed that a GLP-inhibited autophagosome resulted from the decreased of lysosome acidification and lysosomal cathepsin activities, regulated via MAPK/extracelular-signal-regulated-kianse (ERK) activation [76]. The cross-regulation between autophagy and apoptosis is complex and unclear [77]. Pan et al. revealed that the suppression of autophagic flux contributed to GLP-induced apoptosis in CRC cells [76]. Consistently, the inhibition of autophagy promoted cordycepin-induced apoptosis in human neuroblastoma and glioblastoma cells [12].

Autophagy plays a complex role in tumour development and progression; compounds from natural plants or microbes are important resources for drugs against cancer. Emerging evidence has revealed that medicinal fungi exhibited potent anti-tumour bioactivities by regulating various processes of autophagy, warranting further investigation to serve as potential therapeutic agents for cancer treatment.

### 3.4. Suppression of Angiogenesis

Angiogenesis is the formation of new vessels from an existing vascular network. During tumour progression, the action of angiogenic stimulators surpasses the control of angiogenic inhibitors, allowing the development of unregulated vasculature that helps sustain expanding neoplastic growth [78]. Several pro-angiogenic factors, including vascular endothelial growth factor (VEGF), VEGF receptors, transforming growth factor-β1 (TGF-β1), hypoxia-inducible factor (HIF), and matrix metalloproteinase 2 (MMP-2) are involved in angiogenic processes. Among them, VEGF plays a central role. An extract of *G. lucidum* (containing 13.5% polysaccharides and 6% triterpenes) inhibited capillary morphogenesis (tube formation) of human aortic endothelial cells, and inhibited Erk1/2 and Akt kinase activity, resulting in the suppression of transcription factor activator protein 1 (AP-1), followed by the downregulation of VEGF and TGF-β1 [79]. Consistently, the extract of *C. militaris* exhibited a growth inhibitory effect on melanocytic tumours in a nude mice xenograft model. As evidenced by reduced VEGF and CD 31 expression—demonstrated by immunohistochemistry in extract-treated tumours—the chorioallantoic membrane assay revealed that neovascularisation was significantly inhibited [80]. A purified polysaccharide SP1, extracted from the mushroom *Trametes robiniophila murr* (Huaier in Chinese), demonstrated a significant effect on retarding tumour growth in a xenograft hepatocellular carcinoma (HCC) mouse model. Oral administration of SP1 suppressed microvessel density formation in BALB/c nude mice tumour tissues, decreased serum MMP-2, and downregulated the expression of HIF-1α and VEGF, suggesting the promising inhibitory effect of SP1 on angiogenesis [81]. Although the underlying mechanisms require further study, it hinges on the potential role of mycomedicine in anti-angiogenic treatment and provides an alternative strategy for cancer therapy.

### 3.5. Reduction in Metastatic Potential

Metastasis, in which cell adhesion, migration, and invasion are critical steps, plays a vital role in cancer. Numerous regulatory molecules are involved in this complex progression, including cadherin, Snail, focal adhesion kinase (FAK), and MMPs; regulation of these molecules may contribute to the suppression of cancer metastasis [82].

Epithelial-mesenchymal transition (EMT) plays a fundamental role in regulating tumour invasion and metastasis. One hallmark of EMT is the downregulation of epithelial-like markers (E-cadherin, γ-catenin, etc.) to reinforce the destabilisation of adherent junctions, as well as the upregulation of mesenchymal-like markers (N-cadherin, vimentin, etc.) [83]. The transcription factors of EMT regulators, such as Slug, Snail, and Twist, mediate the repression of the epithelial phenotype and are activated in the early phase of the EMT process [84]. For example, recombinant Ling Zhi-8 (rLZ-8), a recombinant protein of medicinal fungus *G. lucidum*, significantly decreased the lung and liver metastatic nodules in a mouse lung cancer metastatic model established by tail vein injection with LLC1 cells and inhibited the mobility of non-small-cell lung cancer (NSCLC) cells in vitro. EMT progression was suppressed by rLZ-8, and protein levels of N-cadherin, vimentin, and Slug were decreased, whereas those of γ-catenin and E-cadherin were increased, both in vivo and in vitro. Mechanistically, the Slug protein was degraded via the MDM2-mediated ubiquitination proteasome pathway, owing to rLZ-8-induced FAK inactivation. Furthermore, rLZ-8 was found to bind to filopodia and interfere with cell adhesion [85]. Consistently, *P. cocos* combined with oxaliplatin exhibited significant inhibitory capacity against gastric cancer cell migration and invasion in both in vitro and in vivo studies. The mRNA and protein expressions of Snail, Twist, vimentin, and N-cadherin were significantly decreased, while the expression of E-cadherin was significantly increased [86].

MMP activities are mostly enhanced in carcinoma cells, promoting the degradation of the basement membrane barrier and extracellular matrix (ECM), allowing direct tumour invasion [87]. MMP-2 and MMP-9 are the two main enzymes of the MMP family, demonstrating abnormally high expression in various malignant tumours. Cordycepin treatment markedly inhibited LNCaP prostate cancer cell metastatic activity, with reduced MMP-2 and MMP-9 protein levels [13]. Further mechanistic studies revealed that inactivation of the PI3K/Akt pathway played an essential role in the anti-metastasis effect of cordycepin. *G. lucidum* extracts (GLE) inhibited the transcriptional level of MMP-9 by downregulating its transcriptional factors, AP-1 and nuclear factor-κB (NF-κB). Moreover, the MAPK/ERK1/2 and Akt pathways were associated with the anti-metastatic activity of GLE in HepG2 liver cancer cells [88].

In addition, multiple interactions during cancer metastasis are mainly mediated by cell-surface adhesion molecules, with the selectin family playing a critical role. All selectins are bound to the cell-surface antigen sialyl Lewis X (sLe^x^), a constituent of the ligand for cell adhesion molecules, which is particularly expressed on the surface of carcinoma cells [89]. Liu et al. reported that a low-molecular-weight polysaccharide isolated from the fruiting bodies of *A. blazei Murill* (LMW-ABP) suppressed the metastasising capacity of cancer cells by inhibiting the interaction between E-selectin and sLe^x^. LMW-ABP effectively inhibited adhesion of HT-29 colorectal cancer cells to human umbilical vein endothelial cells under static conditions, as well as downregulated the expression of sLe^x^ at both the transcriptional and translational levels [90].

### 3.6. Improvement of Immune Response

Apart from directly targeting cancer cells, indirect defence, such as enhancing anti-tumour immunity, plays a crucial role in anti-cancer treatment [91]. Numerous studies have reported immunopotentiation of medicinal mushrooms and their extracts, including lectins, polysaccharides, and proteins with less toxicity but significant immunomodulatory effects [92].

The innate immune system is the body’s earliest barrier against foreign pathogens and plays a vital role in the defence against cancer cells [93]. The bioactive substances derived from mushrooms could activate innate immune cells, including natural killer (NK) cells, T lymphocytes, dendritic cells, and stimulate the expression and secretion of cytokines such as interleukin (IL)-2, tumour necrosis factor (TNF-α), and interferon (IFN-γ) (Figure 2). These cytokines could in turn activate adaptive immunity by stimulating the differentiation of T cells to T helper (Th) cells, which regulate humoral immunity [94]. TP-1, a homogenous polysaccharide extracted from *Huaier* fungus, repressed tumour growth, inhibited pulmonary metastasis, and prolonged the lifetime of hepatic H22 tumour-bearing mice. The relative weights of immune organs and lymphocyte proliferation were improved after TP-1 treatment. Moreover, the percentage of CD4^+^ T cells and NK cells were increased, whereas the number of CD8^+^ T cells was decreased in tumour-bearing mice following TP-1 administration. Furthermore, treatment with TP-1 promoted immune-stimulating serum cytokines, including IL-2 and IFN-γ, but inhibited immune-suppressing serum cytokine IL-10 secretion in H22-bearing mice [95]. Tan et al. revealed that the *Sarcodon imbricatus* extract (SIE) effectively inhibited the growth, migration, and invasion properties of breast cancer cells in vitro and reduced tumour growth in vivo. Mechanically, SIE significantly increased serum concentrations of IL-2, IL-6, and TNF-α, as well as NK cell activity and the viability of splenocytes, while decreasing the expression of programmed cell death-ligand 1 (PD-L1) in 4T1 tumour-bearing mice [96].

Regulatory T (Treg) cells are a subset of CD4^+^ T cells that control autoimmune responses and maintain immune homeostasis. In tumour immunity, Treg cells are involved in tumour development and progression by secreting immunosuppressive cytokines, such as TGF-β, to suppress cancer cell-specific immune reactions [97,98]. Tanaka’s group revealed that oral ingestion of a *Lentinula edodes* mycelia (LEM) extract could inhibit tumour growth following the subcutaneous inoculation of B16 melanoma in C57BL/6 mice, which was not observed in nude mice, suggesting a T cell-dependent mechanism for LEM. After treatment with LEM extract, the percentage of Treg (CD4^+^ Foxp3^+^) cells in the spleen, the mRNA level of Treg cell marker Foxp3 in melanoma tissues, and the level of plasma TGF-β were all significantly reduced. These findings suggested that LEM extract could induce anti-tumour effects possibly via mitigation of Treg-mediated immunosuppression [99].

Collectively, the immunomodulatory effects of medicinal fungi have been well proven, indicating that mycomedicine could be developed and employed as a cancer immunotherapy agent in the future.

### 3.7. Regulation of Oxidative Stress

Damage to DNA, proteins, and lipids, caused by oxidative stress, as well as the resulting elevated reactive oxygen species (ROS), are important contributors to the formation and progression of cancer [100]. However, cancer cells are more sensitive to an acute increase in ROS than non-malignant cells, and excessive ROS levels can also cause a redox imbalance and toxicity in cancer cells [101]. Intriguingly, researchers have observed that medicinal mushrooms demonstrated dual functions in regulating oxidative stress, including both antioxidant and pro-oxidant properties, which provided a potential cancer treatment. For example, a solution of *G. Lucidum*, containing 6% of triterpenes and 13.5% of polysaccharides (GLC), co-administered with chemotherapeutic agent 5-fluorouracil (5-FU) inhibited CRC cell growth and invasive behaviour both in vitro and in vivo. GLC induced oxidative DNA damage and increased ROS formation in CRCs, whereas it protected non-malignant cells from ROS accumulation concurrently. The accumulated oxidative DNA damage further enhanced the sensitisation of cancer cells to 5-FU, resulting in improved anti-cancer effects of 5-FU [102]. Further structure–activity relationship studies revealed that the accumulation of oxidative DNA damage positively correlated with the degree of acetylation in the *G. lucidum* triterpenoid structure, largely owing to the enhanced cell membrane penetrating ability [11].

Moreover, aqueous and ethanolic extracts of *G. lucidum* suppressed human ovarian cell growth and induced antioxidant/detoxification activity, and hence, was suggested as an adjunct supplementary agent in chemotherapy. Mechanically, the chemopreventive activities elicited by *G. lucidum* were attributed to the induction of antioxidant enzymes (superoxide dismutase (SOD) and catalase (CAT)) and activation of phase II detoxification enzymes (NAD(P)H quinone dehydrogenase 1 (NQO1) and glutathione S-transferase pi gene (GSTP1)) via elicitation of the nuclear factor erythroid 2-related factor 2 (Nrf2)-mediated signalling pathway [103]. Similarly, oral administration of *A. Blazei* polysaccharides in breast cancer-bearing rats induced significant tumour growth inhibition by decreasing the levels of lipid peroxidation products and promoting the activities of antioxidant enzymes (SOD, CAT, glutathione peroxidase (GPx), and glutathione reductase (GR)) in the blood and liver. Mechanistically, increased SOD activity improves the efficiency of superoxide anion dismutation into hydrogen peroxide, whereas increased CAT activity detoxifies hydrogen peroxide and converts lipid hydroperoxides to nontoxic substances [104]. Anita et al. reported that H_2_O_2_-induced migration of MCF-7 breast cancer cells was inhibited by *G. lucidum* through the marked inhibition of lipid peroxidation and phospho-Erk1/2 protein levels, which resulted in the downregulation of c-fos expression and inhibition of transcription factors AP-1 and NF-κB [105].

Based on these results, we propose that mycomedicine may represent a promising supplement to regulate oxidative stress, not only destroying cancer cells by inducing oxidative damage, but also preventing normal cells from cancerisation caused by exposure to excessive oxidative stress.

### 3.8. Regulation of Gut Microenvironment

Gut microbiota is essential for human health, which could regulate the host immune system and protect against pathogens through multiple processes. An imbalance in the gut microenvironment causes dysbiosis of the gut microbiota and subsequently leads to various disorders and diseases, including gastrointestinal tract disorders and different types of cancers [106]. Among the large amounts of bioactive macromolecules, mushroom polysaccharides are one of the most well known in regulating the community of gut microbiota [107]. Numerous studies have shown that medicinal mushrooms can modulate the gut microenvironment and simultaneously present anti-cancer properties [108]. For example, GLP suppressed tumour growth in breast cancer-bearing mice and shaped the composition of gut microbiota by elevating the relative abundance of beneficial bacteria, such as Firmicutes and Proteobacteria, while reducing the number of harmful ones, such as Actinobacteria, Bacteroidetes, and Cyanobacteria [109]. Consistently, Luo et al. observed that GLP regulated the gut microbiota by increasing the abundance of Bacteroides, Parabacteroides, Peptostreptococcaceae, Enterobacteriaceae, and Sutterella and decreasing that of Akkermansia, Desulfovibrionaceae, and Ruminococcus in azoxymethane (AOM) and dextran sodium sulphate (DSS)-induced CRC cancer Balb/c mice. More importantly, the tumour numbers and sizes and CRC-related genes were all significantly reduced after treatment with GLP [110]. However, current studies merely describe the correlation between bacteria modulation by mycomedicine and anti-cancer outcomes, with the exact mechanisms largely unknown. Therefore, as a modulator of the gut microbiota, more in-depth studies are urgently required to completely elucidate the multiple benefits and underlying mechanisms of action of mycomedicine.

### 3.9. Reversion of Multidrug Resistance

Multidrug resistance (MDR), defined as cancer cells gaining resistance to chemotherapeutics via different mechanisms, is one of the most significant causes of cancer treatment failure [111]. Aberrant overexpression of the ATP-binding cassette (ABC) transporter family enhances the efflux of anti-cancer agents from cancer cells, resulting in a low intracellular drug concentration. MDR-associated proteins (MRPs) belong to subfamily C of the ABC transporter superfamily and have emerged as an important contributor to chemoresistance. Therefore, blocking or inactivating transmembrane proteins, such as multidrug resistance protein 1 (MDR1, also known as P-gp and ABCB1) and MDR-associated protein 1 (MRP1/ABCC1), provides potential cancer therapies targeting MDR [112,113].

Doğan et al. demonstrated that the methanol extract of *Fomes fomentarius* and ethanol extract of *Tricholoma anatolicum* significantly reversed MDR in breast cancer cells by inhibiting P-gp activity [114]. Clitocine, a natural compound isolated from *Leucopaxillus giganteus*, revealed cytotoxicity against doxorubicin-induced resistance in human hepatocellular carcinoma R-HepG2 cells and uterine carcinoma MES-SA/Dx5 cells. Clitocine significantly downregulated the protein level of P-gp, suppressing MDR1 mRNA levels and promoter activity, accompanied by increased sensitivity and cellular accumulation of doxorubicin in R-HepG2 cells. Further studies showed that the transcription factor NF-κB was involved in the suppression of MDR1 mediated by clitocine. Mutation at the NF-κB binding site or overexpression of NF-κB p65 significantly reversed the inhibitory effect of clitocine on P-gp in R-HepG2 cells, with consistent results observed in the xenografted nude mice model following clitocine treatment [17].

Moreover, the activities of *Ganoderma* and its extracts on the reversal of tumour cell MDR have been widely reported. Preincubation with a water extract of *G. lucidum* in multidrug-resistant small-cell lung cancer cells significantly reduced the IC_50_ of etoposide and doxorubicin, respectively [115]. A fungal protein GMI inhibited P-gp overexpressing docetaxel resistant A549/D16 tumour growth in a xenograft mouse model and elevated the intracellular calcium level to sensitisation of MDR sublines to cell death. Furthermore, apoptosis and autophagy in MDR sublines were differentially enhanced following GMI treatment through the Akt-mTOR-p70S6K pathway [116].

Mycomedicine could serve as natural inhibitors of P-gp and MRP-1 to reverse MDR, which are less toxic than synthetic agents. However, current studies regarding the anti-MDR effects of mycomedicine are primarily focused on reducing drug efflux. Determining whether mycomedicine could demonstrate other anti-MDR mechanisms, such as boosting drug uptake and decreasing drug metabolism [112], is of great importance to accelerate its development for anti-MDR strategies.

## 4. Clinical Evidence of Mycomedicine in Cancer Therapy

The anti-cancer effectiveness of mycomedicine, including *C. sinensis, C. versicolor, G. lucidum*, *G. frondosa*, etc., have been clinically tested on different stages of cancer patients [117]. Of the outcomes assessed in the clinical trials on different stages of cancer, improvement of quality of life (QOL) is the prominent treatment outcome of mycomedicine. In 48 patients with unresectable CRC, a 12-week course of oral daily administration of superfine dispersed lentinan (SDL) (a polysaccharide purified from *L. edodes*) significantly improved the relative QOL scores in nearly halves of the patients [18]. Lower mortality rates and longer survival times were also observed in advanced cancer patients when treated with *Antrodia cinnamomea* for 6 months, compared with those receiving placebo [118]. Besides the above effects, mycomedicine improves the cellular immunological function and assists in maintaining the immune balance. Jinshuibao capsules (JSBC), a Chinese patent medicine capsule mainly consisting of *C. sinensis*, strengthened the immunity and ameliorated the QOL of 36 advanced cancer patients [119]. Treatment with Ganopoly, a *G. lucidum* polysaccharide extract, evidently improved the immune response of 34 advanced-stage cancer patients, resulting in increased of IL-2, -6, IFN-γ and NK activities, and reduced mean plasma concentrations of IL-1 and TNF-α [120]. The clinical efficacy of D-Fraction, a *G. frondosa β*-glucan extract, was observable in 10 patients with early and advanced stage of lung, breast, lingual or gastric cancer. The D-Fraction treatment clearly reduced the levels of tumor markers, including carcinoembryonic antigen (CEA), carbohydrate antigen 15-3 (CA15-3), and carbohydrate antigen 19-9 (CA19-9). Furthermore, the activity of NK cell that involves in suppressing cancer progression, was intensified in all patients [121].

More importantly, the potential of medicinal mushroom-derived pharmaceutical agents as an adjuvant in conventional cancer treatment and synergistically improve the outcome of cancer patients, including QOL, survival duration and regulation of immune system. In concordance with the preclinical findings of LEM in boosting the immune response [99], co-administering LEM with chemotherapy in different types of cancer patients substantially improved their QOL and NK cell activity, as well as decreased their immunosuppressive acidic protein (IAP) levels, compared with patients receiving chemotherapy alone [122]. Similarly, in post-surgery breast cancer patients, combining LEM extract with estrogen-targeted hormone therapy effectively improved their QOL and immune function [123]. Numerous clinical studies have been conducted in China on various types of cancer patients, providing evidence that combinatorial chemotherapy–lentinan treatment showed stronger effectiveness and response rates than chemotherapy alone. The examined types of cancer included lung, gastric, colorectal and gynecology cancer [124]. A meta-analysis on 13 randomised control trials highlighted that adding *C. versicolor* significantly lowered the risk of 5-year mortality by 9% in the treated cancer patients compared to conventional cancer treatment alone. The survival benefit was more pronounced in breast, gastric and colorectal cancer patients [125].

To date, several clinical trials are examining the anti-cancer effect of *Ganoderma*-related compounds including *Ganoderma* spore for NSCLC (phase II) and gastrointestinal neoplasms (phase III), as well as *G. lucidum* extract for breast cancer (Phase I/II) [126]. These ongoing clinical trials are expected to corroborate previous clinical evidence on anti-tumor potential of mycomedicine-derived compounds and facilitate its development as anti-cancer agent.

## 5. Discussion

Medicinal uses of traditional natural products have long been applied but have been primarily dependent on ancient experience and evidence in eastern countries, especially in China, for thousands of years. Since the development of the famous fungal extract, penicillin, and the notable 2015 Nobel Prize for the discovery of artemisinin from traditional Chinese herbs, a growing body of research has acknowledged the potential pharmacological and beneficial effects of these natural-based products. Among them, mycomedicine, which consists of all macroscopic fungi, medicinal mushrooms, and their extracts or powders, contributes an enormous source of food and health supplements for humans, with presents numerous benefits, including anti-cancer, anti-bacterial, and anti-inflammatory properties. As discussed in this review, numerous edible and medicinal mushrooms (Table 1) containing various kinds of bioactive components have demonstrated significant anti-cancer activities (Table 2, Figure 2).

Despite promising advantages, several obvious limitations need to be resolved. First, aside from the widely documented anti-cancer components of mycomedicine, the exact structures and anti-neoplastic functions of numerous bioactive complexes and extracts have not been extensively explored. Therefore, rigorous chemical analyses are urgently required to identify their constituents, as well as the relationship between their structures and anti-cancer activities. More importantly, the amount of these pharmaceutically active compounds isolated from mycomedicine is usually extremely low, with extraction procedures being costly and time-consuming. In this regard, improving the extraction methods and purification protocols could be a practical strategy to enhance production. Moreover, designing and developing organic synthetic routes for obtaining these active substances appears to be another convincing approach. Lastly, although medicinal mushrooms are superior in terms of their safe application and less severe side effects to humans, their efficacy is usually lower than that of synthetic agents. This vital weakness leads to the phenomenon that mycomedicine is consistently regarded as an adjuvant rather than a main therapeutic approach for treating diseases, including cancers.

Although the preclinical and clinical evidence on anti-tumor bioactivities of mycomedicine, medicinal mushrooms-derived anti-cancer agents are currently clinically unavailable or not approved by the US Food and Drug Administration (FDA). The primary obstacles in the clinical development of medicinal mushrooms include insufficient elucidation of both the anti-cancer bioactivities and the identity of the active components. These obstacles are largely result from the considerable gaps between laboratory conditions and clinical applications, including differences in species homology, varied bioavailability between animals and humans, potential off-target issues, and the imbalance of the immunosuppressive tumor microenvironment. However, these gaps still contribute to finding suitable biomarkers amongst the complicated anti-tumour signaling pathways and evaluating the anti-cancer mechanisms of mycomedicine in more detail. Further appropriate animal models and clinical trials are required to obtain more precise information, including the determination of clinical indication, pharmacokinetics and pharmacodynamics profiles, of mycomedicine in humans. Nevertheless, as a tremendous reservoir of pharmacologically active chemical compounds, mycomedicine deserves further development, and in-depth studies should be conducted to reveal the mechanisms of action of this “superfood” in treating various diseases, especially cancers.

## Figures and Tables

**Figure 1 molecules-26-01113-f001:**
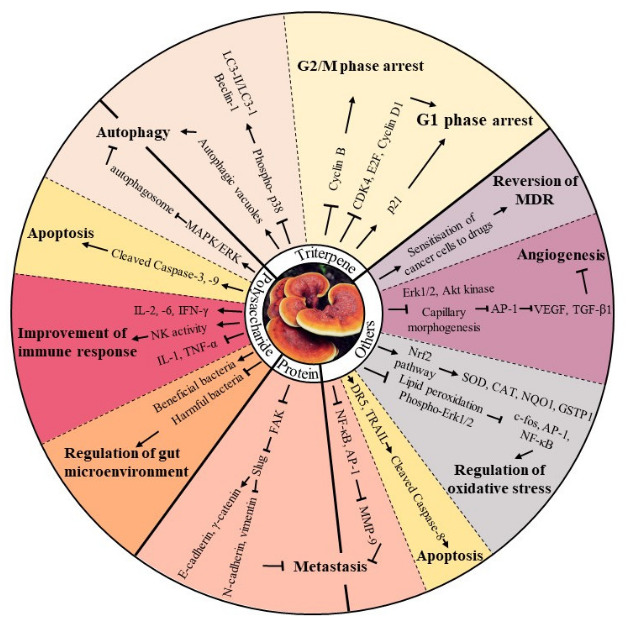
An overview of the numerous anti-tumour bioactivities of *Ganoderma lucidum*. Pharmaceutically active components isolated from *Ganoderma lucidum* demonstrated potent anti-tumour bioactivities through multiple signalling pathways. “↑” represents activate, promote, stimulate or up-regulate, while “Т” represents inhibit, suppress, decrease or down-regulate.

**Figure 2 molecules-26-01113-f002:**
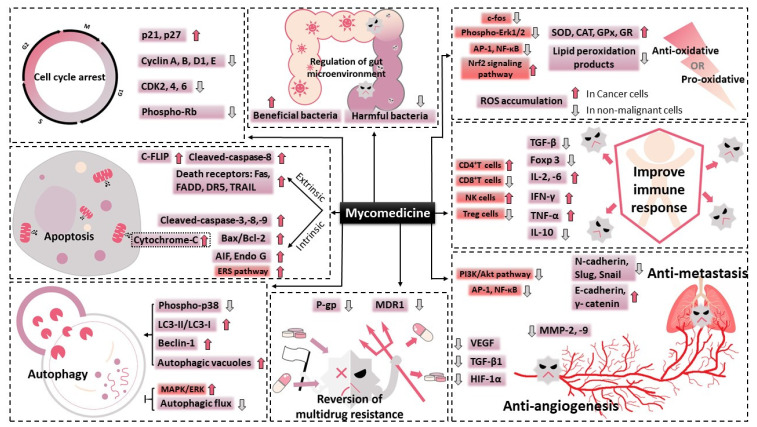
The schematic diagram of the anti-tumour bioactivities of mycomedicine. Mycomedicine demonstrated anti-cancer activities through regulation of cell cycle, apoptosis, autophagy, angiogenesis, oxidative stress, metastasis, multidrug resistance and gut microenvironment. “↑” indicates activate, promote, stimulate or up-regulate, while “↓” and “T” indicates inhibit, suppress, decrease or down-regulate.

**Table 1 molecules-26-01113-t001:** Scientific classification and common name of mycomedicines discussed in this review.

Species	Genus	Family	Order	Class	Phylum	Kingdom	Synonyms	Common Names
*Ganoderma lucidum* (Curtis) P. Karst.	*Ganoderma*	*Ganodermataceae*	Polyporales	Agaricomycetes	Basidiomycota	Fungi	*Boletus lucidus* Curtis *Polyporus lucidus* (Curtis) Fr.	Lingzhi (In Chinese) Reshi mushroom (In Japanese)
*Ganoderma tsugae* Murrill							*Fomes tsugae* (Murrill) Sacc. and D. Sacc. *Polyporus tsugae* (Murrill) Overh.	Hemlock varnish shelf
*Ganoderma microsporum* R.S. Hseu							N.A.	N.A.
*Grifola frondosa* (Dicks.) Gray	*Grifola*	*Grifolaceae*					*Boletus frondosus* Dicks. *Polyporus frondosus* (Dicks.) Fr.	Huishuhua (In Chinese) Maitake (In Japanese)
*Trametes versicolor* (L.) Lloyd	*Trametes*	*Polyporaceae*					*Boletus versicolor* L. *Coriolus versicolor* (L.) Quél.	Yunzhi (In Chinese) Turkey tail (Common name)
*Trametes robiniophila* Murrill							*Perenniporia robiniophila* (Murrill) Ryvarden *Poria robiniophila* (Murrill) Ginns	Huaier (In Chinese)
*Fomes fomentarius* (L.) Fr.	*Fomes*						*Agaricus fomentarius* (L.) *Lam.* *Boletus fomentarius* L.	Tinder fungus (Common name) Hoof fungus (Common name) Ice man fungus (Common name)
*Macrohyporia cocos* (Schwein.) I. Johans. and Ryvarden	*Macrohyporia*						*Poria cocos* (Schwein.) F.A. Wolf *Sclerotium cocos* Schwein.	Fuling (In Chinese) Hoelen (Common name)
*Lentinus squarrosulus* Mont.	*Lentinus*						*Pleurotus squarrosulus* (Mont.) Singer *Lentinus crenulatus* Massee	N.A.
*Agaricus blazei* Murrill	*Agaricus*	*Agaricaceae*	Agaricales				N.A.	Jisongrong (In Chinese) Himematsutake (In Japanese) God’s mushroom (Common name)
*Lentinula edodes* (Berk.) Pegler	*Lentinula*	*Omphalotaceae*					*Collybia shiitake* J.Schröt. *Agaricus edodes* Berk.	Xianggu (In Chinese) Shiitake (In Japanese) Black mushroom (Common name)
*Hypholoma lateritium* (Schaeff.) P. Kumm.	*Hypholoma*	*Strophariaceae*					*Deconica squamosa* Cooke *Geophila sublateritia* (Fr.) Quél.	Brick cap (Common name) Red woodlover (Common name)
*Schizophyllum commune* Fr.	*Schizophyllum*	*Schizophyllaceae*					*Agaricus alneus* L. *Agaricus multifidus* Batsch	Split gill mushroom
*Hypsizygus marmoreus* (Peck) H.E. Bigelow	*Hypsizygus*	*Lyophyllaceae*					*Clitocybe marmorea* (Peck) Sacc. *Agaricus marmoreus* Peck	Beech mushroom (Common name)
*Leucopaxillus giganteus* (Sowerby) Singer	*Leucopaxillus*	*Tricholomataceae*					*Agaricus giganteus* Sowerby *Paxillus giganteus* (Sowerby) Fr.	Giant leucopax (Common name)
*Flammulina velutipes* (Curtis) Singer	*Flammulina*	*Physalacriaceae*					*Collybia eriocephala* Rea *Agaricus atropes* Schumach.	Velvet shank (Common name) Golden needle mushroom (In Chinese) Enokitake (In Japanese)
*Omphalotus illudens* (Schwein.) Bresinsky and Besl	*Omphalotus*	*Omphalotaceae*					*Clitocybe illudens* (Schwein.) Sacc. *Panus illudens* (Schwein.) Fr.	Eastern jack-o’lantern mushroom (Common name)
*Tricholoma anatolicum* H.H. Doğan and Intini	*Tricholoma*	*Tricholomataceae*					N.A.	Katran Mantari (In Turkish)
*Suillus placidus* (Bonord.) Singer	*Suillus*	*Suillaceae*	Boletales				*Gyrodon fusipes* (Fr.) Sacc. *Gyrodon placidus* (Bonord.) Fr.	Slippery white bolete
*Sarcodon imbricatus* (L.) P. Karst.	*Sarcodon*	*Bankeraceae*	Thelephorales				*Hydnum imbricatum* L. *Sarcodon gracilis* (Fr.) Quél.	Shingled hedgehog (Common name) Scaly hedgehog (Common name)
*Cordyceps militaris* (L.) Fr.	*Cordyceps*	*Cordycipitaceae*	Hypocreales	Sordariomycetes	Ascomycota		*Clavaria granulosa* Bull. *Hypoxylon militare* (L.) Mérat	N.A.
*Ophiocordyceps sinensis* (Berk.) G.H. Sung, J.M. Sung, Hywel-Jones and Spatafora	*Ophiocordyceps*	*Ophiocordycipitaceae*					*Cordyceps sinensis* (Berk.) Sacc. *Sphaeria sinensis* Berk	Dongchongxiacao (In Chinese) Yartsa gunbu (Common name)
N.A. represents not available								

**Table 2 molecules-26-01113-t002:** The structure and anti-cancer action of bioactive compounds isolated from mycomedicines that are included in this review.

Species	Compound Name	Structure	Anti-tumour Bioactivity	Reference
*Agaricus blazei*	Agaritine	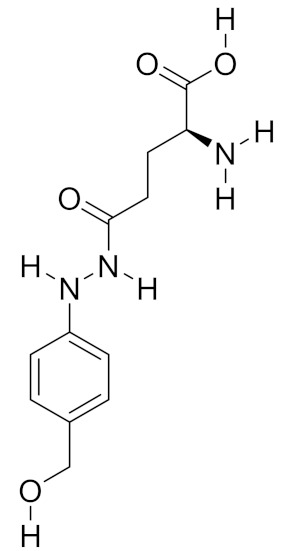	Induction of intrinsic apoptosis	[9]
*Ganoderma lucidum*	Ganoderic acid DM	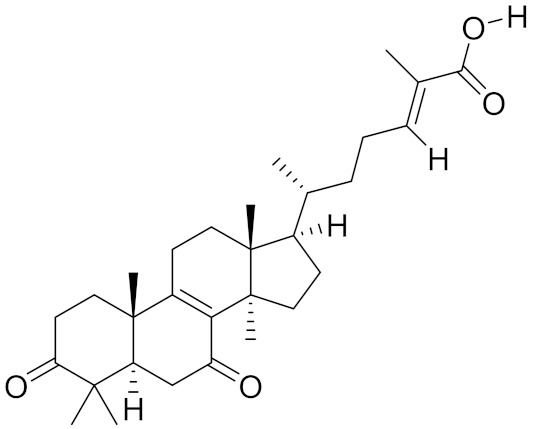	Induction of G1-phase cell cycle arrest; induction of apoptosis	[10]
	Ganoderic acid T	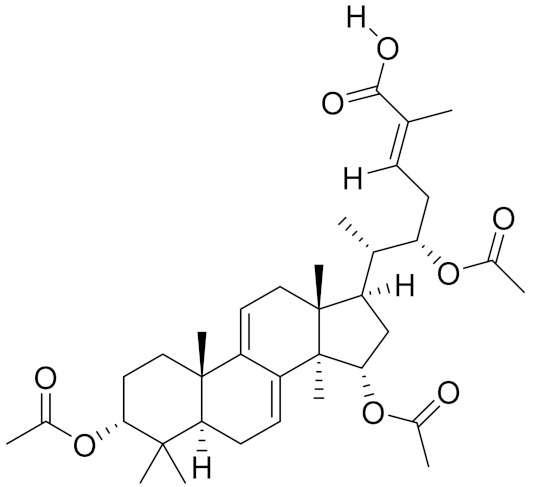	Induction of apoptosis; regulation of oxidative stress	[11]
*Cordyceps militaris*	Cordycepin	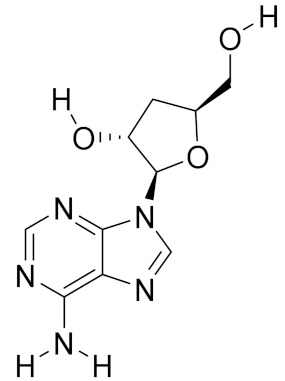	Induction of S-phase cell cycle arrest; inhibition of autophagy; reduction in cancer metastasis	[12,13,14]
*Suillus placidus*	Suillin	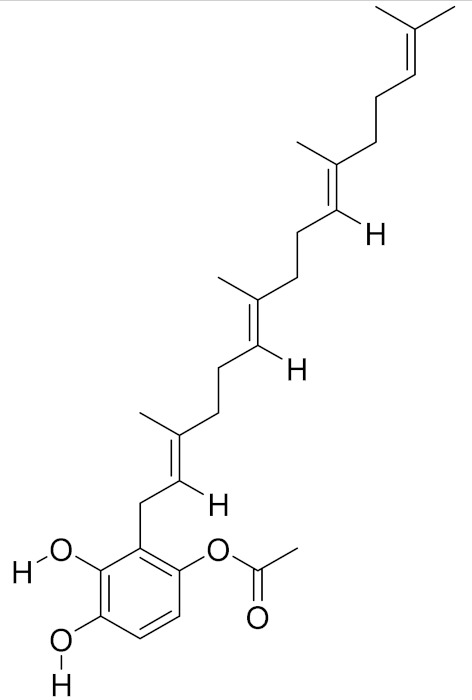	Induction of extrinsic apoptosis	[15]
*Poria cocos*	Poricotriol A	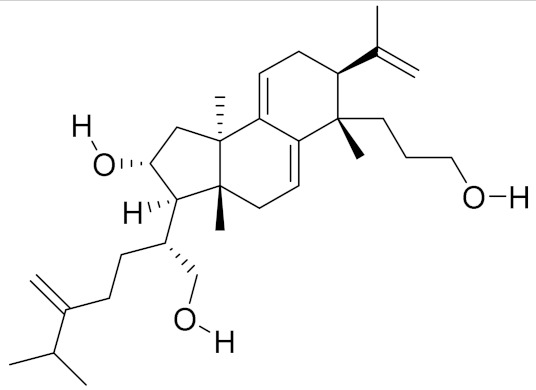	Induction of intrinsic apoptosis through caspase-independent pathway	[16]
*Leucopaxillus giganteus*	Clitocine	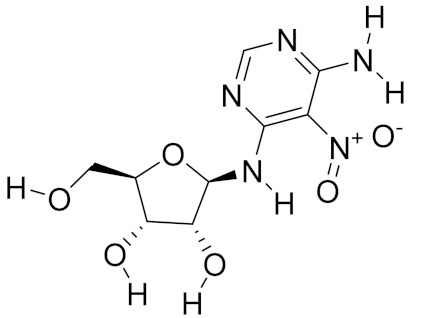	Reversion of multidrug resistance (MDR)	[17]
*Lentinula edodes*	Lentinan	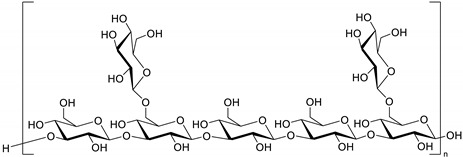	Improvement in quality of life of cancer patients	[18]

## Data Availability

No new data were created or analyzed in this study. Data sharing is not applicable to this article.

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
