# Peer review of "Mycomedicine: A Unique Class of Natural Products with Potent Anti-tumour Bioactivities"

_molecules, 2021, doi:10.3390/molecules26041113_

Round 1
Reviewer 1 Report
Authors in this review provide an exhaustive description of about 20 compounds isolated from edible mushrooms and medicinal fungi, a class of compounds included in the Mycomedicine, which have exhibited promising anti-tumor activities. In particular, they well describe the characteristics of the active components isolated from the Mycomedicine including polysaccharides, terpens and terpenoids and proteins. In addition, they describe the mechanisms underlying the anti-cancer activity shown by these components.
To the best of my knowledge, this type of review is not available in the literature to date. The review is well written and can be accepted in this form. Moreover, in my opinion the information reported in this review can be of interest to a broad Molecules audience."
Reviewer 2 Report
Dear Authors,
I have reviewed the manuscript titled "Mycomedicine: A Unique Class of Anti-Cancer Natural Products", which consists of a review of the published literature regarding the effects of some compounds obtained from fungi, against different molecular targets involved in the complex pathology called Cancer.
The introduction is short, but makes the topic clear. Table 1 shows interesting data.
Table 2 needs to be improved, since the size of the letters of the heteroatoms is not adequate, and the resolution of structure figures is not optimal. It is also suggested that the sizes of the different structures be homogeneous throughout the table.
The description of the polysaccharides is well done, the different types of polysaccharides that have been previously obtained were adequately reviewed. However, no structure of this type of bioactive compound was included in Table 2.
On the other hand, the description of terpenes and terpenoids in section 2.2 contains some flaws in the way of describing this group of compounds. For example, the phrase "Structurally, triterpenes contain six isoprene units, which can form linear chains or ring-like structures" is not appropiate to the description of this important class of compounds.
Another example is the phrase "with chemical structures consisting of four cyclic and two linear isoprenes", which is erroneous, since these types of compounds do not present two isoprene units in linear form.
Another error in the way to describe terpenes is the phrase "Sesquiterpenes, consisting of three isoprene units, are another major class of terpenes extracted from mycomedicinal agents, often demonstrating the molecular formula C15H24", which is not correct, since sesquiterpenes generally contains 15 carbon atoms, but the number of hydrogens depends on the number of unsaturations, and the molecular formula of a group of compounds cannot be generalized in this way.
In the other sections, the authors limit to describing some examples of compounds that exhibit certain biological activities, mainly in cancer cell lines, but the manuscript has no evidence of having been done in a comprehensive and systematic manner as established in the introduction.
I consider that the main problem with the manuscript is that the authors confuse the term "anti-cancer" with the term "cytotoxicity" or with the term "bioactivity". In the study, only the reference [86] describes a clinical study, and the authors end their discussion with the phrase "Currently, most research is conducted under laboratory conditions, and further appropriate animal models and clinical trials are required to obtain more precise information, including clinical indication confirmation and pharmacokinetics and pharmacodynamics profiles, of mycomedicine in humans ", which implies that the compounds and extracts described in the manuscript do not be considered as anti-cancer.
My recommendation is to reject, in order to improve the manuscript.
Round 2
Reviewer 2 Report
Dear Authors,
Thank you for considering my recommendations.
I have read with pleasure the new version of his manuscript, which has been substantially improved.
My only comment for this new version is the following:
In Table 2, examples of bioactive compounds that have been isolated from different organisms are described, such as: agaritin (an amino acid), cordycepin (a purine) and clitocine (a ribofuranosylamino-pyrimidine), but them are not described in the section 2 Bioactive Components Isolated from Mycomedicine.
Since much of the manuscript focuses on describing the biological activities of these compounds, I suggest adding all the the subsections corresponding to all the compounds described in table 2, and review the chemistry of these compounds as has already been done with polysaccharides, terpenes and terpenoids, and proteins in the same section.
thank you for your attention.
Author Response
Thank you for appreciating our revised work. Concur with your feedback, we have modified the subsection heading “2.3 Proteins” to “2.3 Proteins and amino acids”, and added the relevant contents. To further elucidate other mycomedicine-derived bioactive compounds that are stated in Table 2, we have added a subsection “2.4 Other Bioactive Compounds”. In this subsection, we have discussed nucleosides, phenols and other types of compounds isolated from mycomedicine.
